# Exploring Spatial Mismatch between Primary Care and Older Populations in an Aging Country: A Case Study of South Korea

Jeon-Young Kang [1], Sandy Wong [2], Jinwoo Park [3,*], Jinhyung Lee [4] and Jared Aldstadt [5]

[1] Department of Geography, Kyung Hee University, Seoul 02447, Republic of Korea; geokang@khu.ac.kr
[2] Department of Geography, The Ohio State University, Columbus, OH 43210, USA
[3] Department of Geography and Geographic Information Science, University of North Dakota, Grand Forks, ND 58202, USA
[4] Department of Geography and Environment, Western University, London, ON N6A 3K7, Canada
[5] Department of Geography, University of New York at Buffalo, Buffalo, NY 14260, USA
[*] Correspondence: jpark.geo@gmail.com

**Abstract:** With the rapid growth of aging populations in South Korea, it is important to assess spatial accessibility to healthcare resources as older adults may need frequent visits to hospitals. Healthcare spatial accessibility is measured based on available resources (e.g., physicians, beds, services), demands (e.g., population), and travel costs (e.g., distance or time). In this study, we employed an Enhanced Two-Step Floating Catchment Area (E2SFCA) method to measure the spatial accessibility to primary care for older populations (i.e., aged 65 and older) in major cities in South Korea, including Seoul, Busan, Daegu, Incheon, Gwangju, Daejeon, and Ulsan. We found that the aging population in Seoul, the capital and biggest city in South Korea, has relatively better accessibility than those living in other cities. We also discovered a negative relationship between accessibility to primary care and the aging index (i.e., population over 65 years old/population less than 15 years old); the regions with a higher ratio of older populations have lower accessibility to primary care. The results suggested that more primary care services (perhaps via mobile vans) are needed in regions predominantly with older people to improve their healthcare access.

**Keywords:** aging population; spatial accessibility; two-step floating catchment area method; healthcare access; GIS

## 1. Introduction

South Korea is experiencing rapid growth in its aging population. In addition, there is an increasing demand for its healthcare system, given that older adults have higher rates of health services utilization than younger populations. Currently, people aged 65 years and over comprise 15% of South Korea's total population [1]. This proportion is projected to increase considerably in the future, with forecasts of 39.8% in the year 2050 and 43.9% in the year 2060 [1]. Given this rapidly aging population, various government policies for older adults in South Korea need to be formulated to provide them with sufficient access to healthcare services (e.g., long-term care insurance) [2].

To effectively prepare for future demands on healthcare services with a rapidly growing aging population, it is necessary to evaluate the current state of healthcare service allocation. One critical aspect is to evaluate whether a set of populations (e.g., older people) have sufficient spatial accessibility to healthcare services within urban areas. From the perspective of urban planning, sustainable practices involve enhancing people's local proximity to a broad range of services, including healthcare facilities, social services, parks, and nature [3].

Spatial accessibility is broadly defined as the ease of reaching (and interacting with) desired resources and opportunities and considered the following three variables: demands (e.g., patients), supplies (e.g., beds, physicians, services), and people's capacity to travel.

Measuring people's spatial access to healthcare services would enable a better understanding of geographic variations in healthcare service availability and identification of existing inequities in healthcare accessibility [4]. Also, it can be used as a basis for diagnosing and alleviating spatial mismatches between patients and healthcare services [5,6]. Not only that, spatial accessibility can consider how access to services varies according to the mode of transportation (e.g., automobiles, walking, and public transit [5,7]).

A widely regarded method for measuring spatial accessibility of healthcare services is the Two-Step Floating Catchment Area method (2SFCA) [8]. The 2SFCA method is a spatial analog of the provider-to-population ratios (PPR), which takes into account the estimated demands from surrounding populations and the supplies (e.g., the number of physicians in a clinic) within the range of services. The original 2SFCA method considers whether a location is accessible to a particular service of interest in a Boolean manner (i.e., accessible or not accessible). To resolve this limitation of the conventional approach, there have been advanced developments in the 2SFCA method, including the Enhanced Two-Step Floating Catchment Area (E2SFCA) method [9], the Variable Two-Step Floating Catchment Area (V2SFCA) method [10], and the Modified Two-Step Floating Catchment Area (M2SFCA) method [11]. These methods vary depending on metrics to delineate the catchment areas of hospitals and populations. Specifically, the E2SFCA method takes into consideration the distance decay of catchment areas of populations and hospitals [9]. The V2SFCA method considers the varying catchment areas based on the predetermined physician-to-population ratio [10]. The M2SFCA method employs an additional distance decay function only for supply to validate the assumptions of 2SFCA metrics (i.e., every supply location is optimal for serving demand) [11]. Along with these methodological advancements, spatial accessibility has been investigated among different populations and applied to various contexts (e.g., visual impairment [12], disability [13], food desert [14], transportation mode [15], and COVID-19 [16]).

Due to the rapid growth of older populations in South Korea, much attention has been paid to the spatial accessibility of healthcare services by urban scholars and public health scientists in South Korea. A good example is a paper by Yun et al. [17], which evaluates and compares spatial access to emergency rooms in South Korea via the 2SFCA method based on two different types of populations: census-based and mobile-based. The authors empirically demonstrate the advantages of dynamic population counts from mobile phone data as a more realistic way of representing the demand for emergency rooms, thereby aiding better planning and policy to reduce social inequality in public health. Efforts were also made to consider the unique geographic contexts of South Korean cities when measuring spatial accessibility to medical facilities. For instance, focusing on Seoul, the capital and biggest city in South Korea, Kim et al. [18] developed a new 2SFCA model called the Seoul Enhanced Two-Step Floating Catchment Area (SE2SFCA) method, which considers Seoul's higher population and hospital density compared to North American and European cities. Meanwhile, towards a more general enhancement of 2SFCA, Shin and Lee [19] developed an improved 2SFCA model and applied it to measure spatial accessibility to emergency medical services in South Korea. With the local and global distance decay functions and effective potential demand to handle the under- or over-estimation of potential demand, the improved 2SFCA model provides a more nuanced accessibility measurement result.

Despite the active research on accessibility trends in Korean contexts, the literature has overlooked the intra- as well as inter-urban variations in spatial accessibility to healthcare resources with an explicit focus on aging populations. There are well-known urban-rural inequities in healthcare access [20,21], but much less is known about intra-urban accessibility disparities within South Korean cities [22]. Also, spatial access to healthcare has not been fully investigated with respect to older and aging populations in South Korea. To fill this gap, our paper investigates the spatial accessibility to healthcare services for older populations in metropolitans in South Korea. Older individuals are particularly vulnerable as they are at higher risk of chronic illness (e.g., hypertension and diabetes) and

comorbidities; hence they have a more increased need for health services. In addition, with a distinct geographical distribution of medical services in South Korea (e.g., 30 times-higher hospital density than in the United States [18]), measuring healthcare spatial accessibility in Korean contexts can be a meaningful and insightful case study for other high-density cities across the world.

Our study investigates healthcare spatial accessibility for older populations (aged 65 or over) and disparities across intra- and inter-urban areas, focusing on the seven major cities in South Korea (i.e., Seoul, Busan, Incheon, Daegu, Daejeon, Gwangju, and Ulsan). These cities are projected to have a significant increase in their aging populations in the next twenty years. They were also selected because their metropolitan governments have autonomous rights for decision-making of infrastructure allocation.

The analysis has proceeded in the following three steps. First, we measured spatial accessibility to healthcare resources for the older adult population in the seven cities. This step employed the E2SFCA method [9], given its wide implementation in healthcare accessibility studies [23,24] as well as its intuitive characteristics [25]. In addition, we explored different transportation modes (i.e., driving and walking). Second, we compared the accessibility characteristics between cities and examined the disparity of access within the cities. Third, we conducted Kendall's tau correlation between the aging index and spatial accessibility. This study answers the following three research questions: (1) Within cities (intra-urban), which elderly groups and neighborhoods have the lowest and highest spatial access to primary care? (2) Between cities (inter-urban), which ones have the lowest and highest spatial access to primary care? (3) What is the correlation between spatial access to primary care and the aging index? Our findings can help identify populations and areas where policies (e.g., increasing providers in underserved regions) or technologies (e.g., mobile health van, telehealth) are most needed for improving healthcare access, thereby reducing health inequities and long-term healthcare costs.

## 2. Data and Methods

### 2.1. Study Area

Our study area includes the seven major cities in South Korea, which are Seoul, Busan, Incheon, Daegu, Gwangju, Daejeon, and Ulsan. As of March 2020, the total population in South Korea is about 52,000,000. In South Korea, many rural areas have already become super-aged societies. More critically, the seven major cities are expected to become super-aged in the near future [26]. As shown in Figure 1, approximately 15% of the total population is comprised of older adults, and by 2040, the proportion is expected to go up to 35% of the total population.

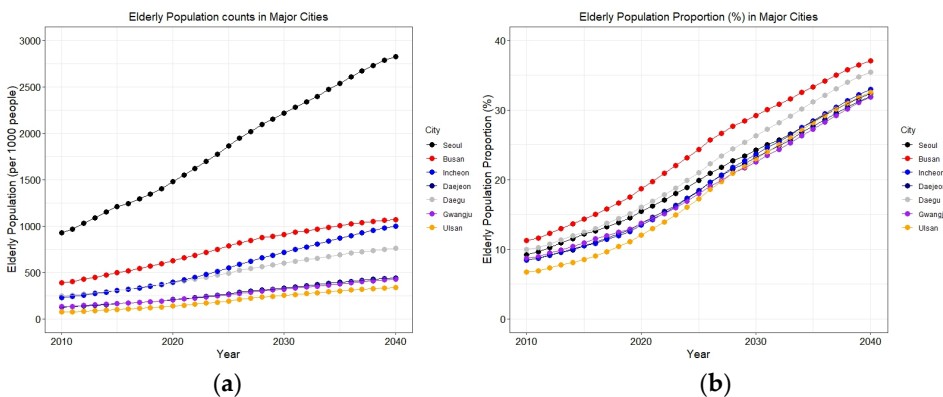

**(a)**　　　　　　　　　　　　**(b)**

**Figure 1.** Aging Population in Major Cities in South Korea. (**a**) Aging population counts (1000 people); (**b**) Elderly population proportion (%).

*2.2. Data*

The following datasets were used: (1) hospital dataset (i.e., the number of physicians at each hospital), (2) residential population information, and (3) road network dataset. We obtained the hospital dataset from the Korea Open Data Portal (http://www.data.go.kr (accessed on 15 March 2023)), which contains ID, type, hospital name, number of physicians, number of rooms, and address. In our analysis, we included only hospitals that provide primary care services. The hospital dataset does not provide the x (longitude) and y (latitude) coordinates of each hospital. To find the x, y coordinates of each hospital, we geocoded all hospitals through KaKao's open Application Program Interface (API) service (https://developers.kakao.com/ (accessed on 15 March 2023)), which is a well-known geospatial information API service in South Korea. The number of hospitals in each city is as follows: (1) 9430—Seoul, (2) 2583—Busan, (3) 1789– Incheon, (4) 1238—Daejeon, (5) 2173—Daegu, (6) 1055—Gwangju, and (7) 559—Ulsan. Figure 2 illustrates the spatial distributions of primary care hospitals in seven cities in South Korea.

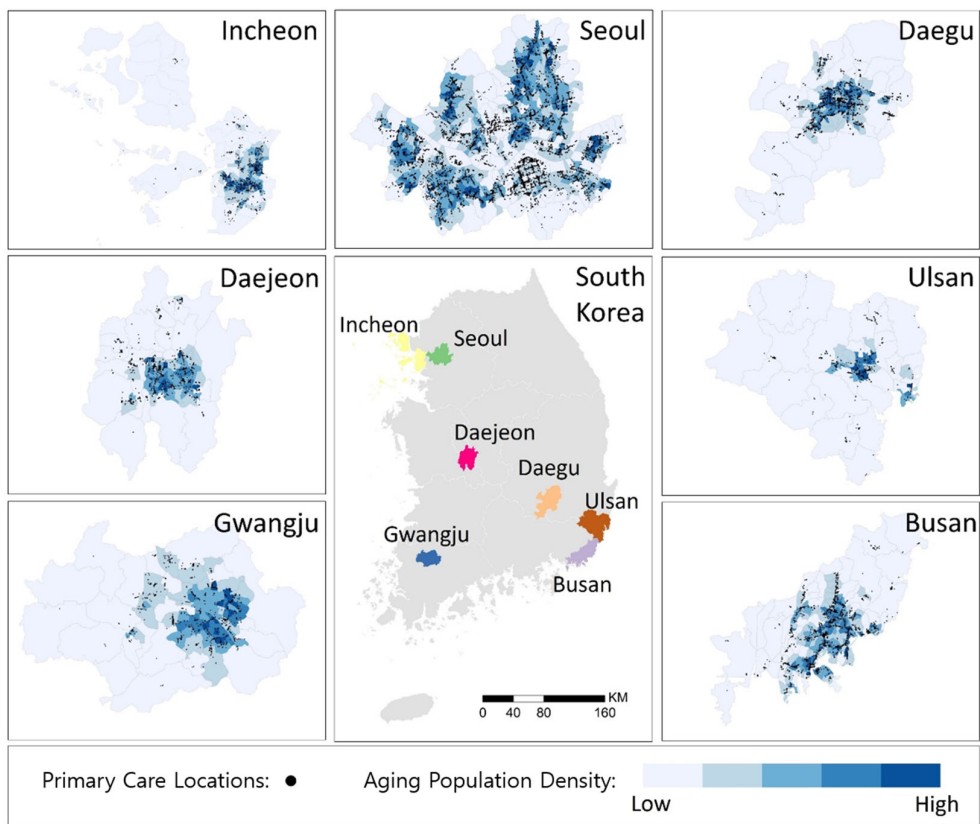

**Figure 2.** Spatial distributions of hospitals in the major cities of South Korea.

The population data came from the Korea Population and Housing Census in 2018 and were obtained from the National Spatial Data Infrastructure Portal (http://www.nsdi.go.kr/ (accessed on 15 March 2023)). At the city level, we extracted data on people who were aged 65 or older to calculate the PPR for each city (Table 1). We also extracted demographic data at the scale of dongs, which are administrative units that are proxies for neighborhoods.

In addition, we obtained the road network dataset (OpenStreetMap data) through a Python package called OSMnx (https://osmnx.readthedocs.io/en/stable/ (accessed on 15 March 2023)). The OSMnx package supports not only download but also analysis of the street network dataset, which is from an open-source map dataset, OpenStreetMap. Using OSMnx, we retrieved two different road network datasets, calculating travel time for driving and walking. As the purpose of this study is not only to analyze the spatial accessibility

to primary care for older populations but also to compare the spatial accessibility measures via driving and walking, we downloaded both road network datasets and utilized them.

**Table 1.** Physician-to-aging population ratio (PPR).

| City | Aging Population | Physician Counts | Physician-to-Population Ratio (per 1000 People) |
|---|---|---|---|
| Seoul | 1,214,965 | 16,868 | 13.88 |
| Busan | 525,167 | 4223 | 8.04 |
| Daegu | 322,602 | 3386 | 10.5 |
| Incheon | 311,151 | 2874 | 9.24 |
| Gwangju | 168,497 | 1808 | 10.73 |
| Daejeon | 166,059 | 2047 | 12.33 |
| Ulsan | 102,803 | 922 | 8.97 |

*2.3. Method*

Spatial accessibility is calculated by the interactions between demand (i.e., counts of the population) and supply (i.e., counts of doctors or beds). To measure spatial access to primary care services for elderly populations, we employed the E2SFCA method [9]. First, it finds the population located at *i* within a hospital's catchment area of each primary care (*j*) and then computes a physician-to-aging population rate $R_j$ within the catchment area. A catchment area is determined by a travel time threshold that the aging population can reach ($d_0$).

$$R_j = \frac{S_j}{\sum_{k \in \{d_{ij} \le d_0\}} P_k}$$

In this study, we measure people's travel times to healthcare providers based on automobile driving and walking times. We assumed that people aged 65 and older would take a cab instead of public transit services (e.g., bus, subway). Except for Seoul, other cities are not fully covered by public transit systems. In addition, we consider walking time to the primary care.

In order to take into account the distance decay effect (i.e., people may be more likely to visit relatively closer primary care hospitals than further ones), we set up three travel time zones: 0–10, 10–20, and 20–30 min for driving and 0–5, 5–10, and 10–15 min for walking. For each travel time zone, we followed Luo and Qi's [9]'s approach and applied the weights of 1, 0.68, and 0.22 to each of the three travel time zones (e.g., a weight of 0.68 for 10–20 driving time and 5–10 walking time), respectively. Then, we made our final calculations using the following formula:

$$A_i = \sum_{j \in (d_{ij} \in D_1)} R_j W_1 + \sum_{j \in (d_{ij} \in D_2)} R_j W_2 + \sum_{j \in (d_{ij} \in D_3)} R_j W_3$$

where $A_i$ is the accessibility of people at location *i* to primary care, and a physician-to-aging population rate $R_j$ at primary care location *j* in which population centroids fall within the catchment area. *W* is the weight for each travel zone. Visualizing the E2SFCA method results allowed us to assess intraurban variations in accessibility.

With the accessibility measures, we conducted three post-analyses to discover intra- and inter-urban disparities in healthcare spatial accessibility for older populations. To be specific, one-way analysis of variance (ANOVA) and Tukey's post hoc analysis were used to compare the spatial accessibility scores among the major cities. The results of ANOVA can indicate whether the mean of accessibility is the same among the cities, and Tucky's post hoc analysis provides the specific differences between each pair of cities. In addition, the Gini coefficients explore spatial inequalities in the geographical access to primary care within each major city [27]. In the extant literature, the Gini coefficient is well-established

as a useful indicator for measuring inequality in spatial accessibility [14,28]. Furthermore, Kendall's tau correlation is employed to assess the mismatch between primary care spatial accessibility and the aging index (i.e., population over 65 years old/population less than 15 years old) per city. Given that we employed the aging index instead of the aging population counts as a correlation variable, it is considered to be exempt from the potential concern of collinearity.

## 3. Results

### 3.1. Measures of Spatial Accessibility

The results from the E2SFCA method show that the spatial accessibility to primary care for older adults varies geographically in all cities (see Figure 3). In Figure 3, darker green colors depict the regions with higher accessibility to primary care for the aging population, while lighter greens represent lower spatial accessibility to healthcare services. In general, given that primary care hospitals are mainly located close to city centers, healthcare accessibility around the center of cities is higher than in periphery regions (i.e., suburbs). Interestingly, Daegu provided distinctive results in driving compared to the other cities. Here, higher accessibility was concentrated in the southeastern part of the city, where there are more socioeconomically advantaged residents.

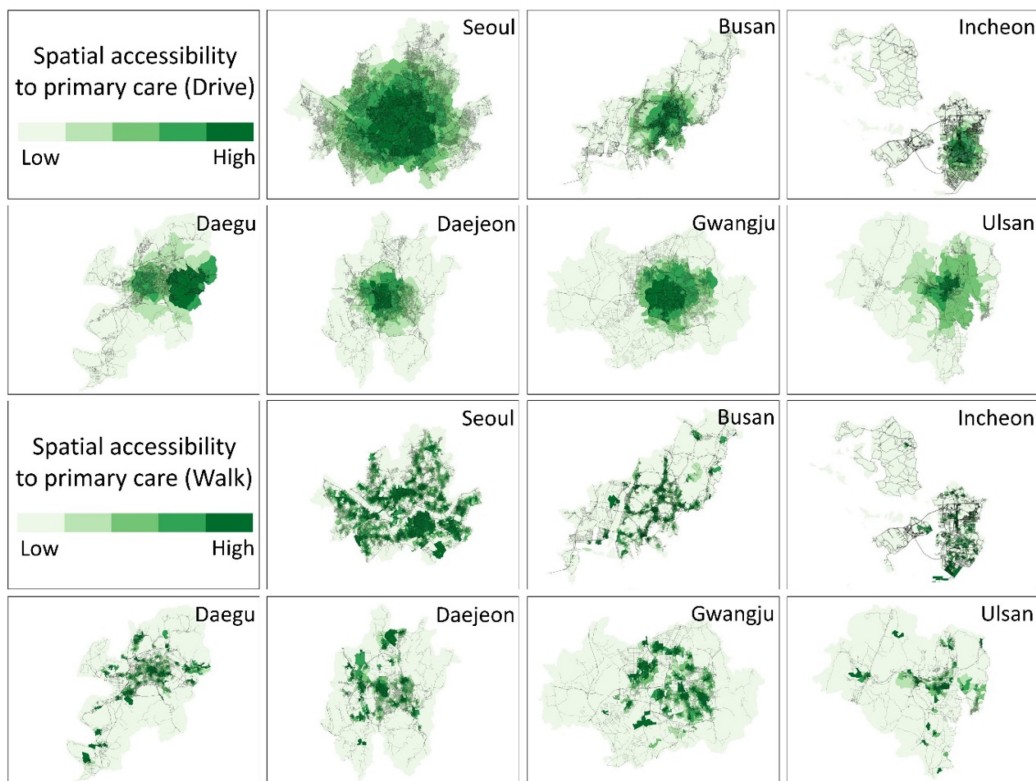

**Figure 3.** Spatial accessibility to primary care for older adults in the major cities in South Korea.

Spatial accessibility by driving had a relatively uniform distribution compared to spatial accessibility by walking (Figure 4). Seoul has the most uniform distribution, while Daegu has the most discrete distribution. Seoul is the most developed city in South Korea, and healthcare infrastructure is concentrated in Seoul. This may explain the relatively even distribution of spatial accessibility in Seoul. In contrast, Daegu has three distinct areas with different levels of accessibility. While the southeastern part of the city has high accessibility, the southwestern and western parts of the city have low accessibility (Figure 3).

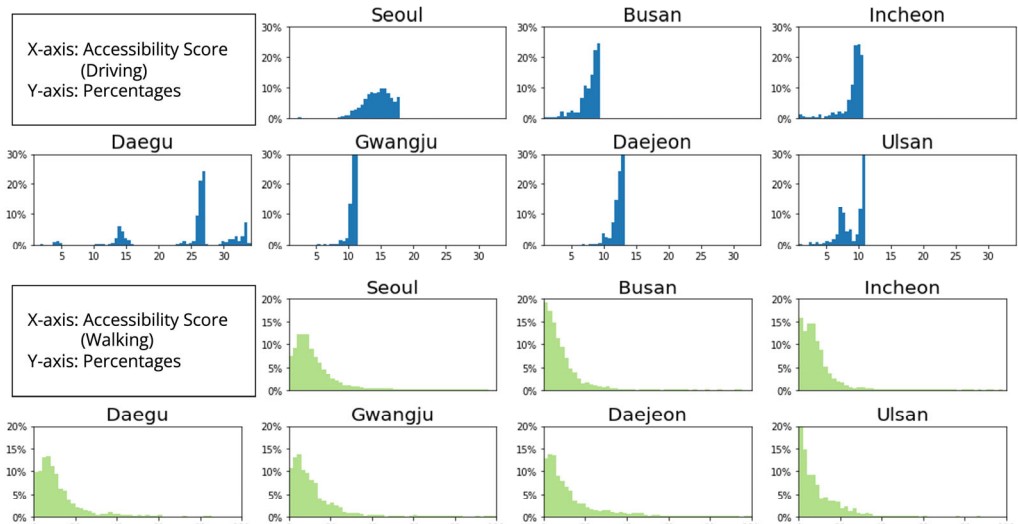

**Figure 4.** Distributions of accessibility measures by either driving or walking in each city.

Spatial accessibility by walking has a right-skewed distribution for every city. This indicates that many subregions within the city have less access to primary care. In other words, strong spatial disparities in healthcare accessibility are detected. Compared to its uniform distribution of driving accessibility, Seoul has a very high accessibility score, with 799.85 as its maximum accessibility score. In summary, spatial accessibility provides a very distinctive pattern between different modalities (i.e., driving versus walking).

### 3.2. Inter-Urban Disparity of Access

The ranks of cities based on physician-to-population ratios do not mirror their ranks based on spatial accessibility. While Seoul has the highest physician-to-population ratio (Table 1), it does not have the highest spatial accessibility. Cities were ranked in a different order based on average measures of spatial accessibility to healthcare resources. Using the average value of accessibility by driving (i.e., number of physicians accessible per 1000 older adults), the cities are ranked as follows: Daegu (25.61), Seoul (14.40), Daejeon (12.31), Gwangju (10.72), Incheon (9.12), Ulsan (8.92), and Busan (8.00). The average accessibility values with walking rank the cities in a different order: Seoul (14.16), Daejeon (11.65), Gwangju (10.78), Daegu (10.73), Ulsan (9.14), Incheon (9.01), and Busan (8.38). In summary, Ulsan, Incheon, and Busan are the cities that have limited healthcare resources (PPR) as well as insufficient spatial accessibility by both walking and driving.

To quantify the statistical difference in spatial accessibility among cities, we performed a one-way ANOVA and Tukey's post-hoc analysis. We found statistically significant differences in accessibility to primary care by driving among cities ($F_{(45,532, 6)} = 22,313$, $p < 0.001$). According to Tukey's post-hoc analysis results, the spatial accessibility between cities was statistically different ($p < 0.001$), except for the cases of Ulsan and Incheon. Also, we found statistically significant differences in spatial accessibility to primary care by walking among cities ($F_{(45,532, 6)} = 102.8$, $p < 0.001$). The results from Tukey's post-hoc analysis showed that spatial accessibility was not different between Gwangju and Daegu, Seoul and Daejeon, and Ulsan and Incheon.

In addition, spatial accessibility in each city was different depending on the mode of transportation (i.e., driving versus walking): Seoul ($t_{(19,466)} = -4.5286$, $p < 0.001$), Busan ($t_{(7110)} = -6.9685$, $p < 0.001$), Incheon ($t_{(6037)} = -6.6472$, $p < 0.001$), Daegu ($t_{(7432)} = 58.863$, $p < 0.001$), Gwangju ($t_{(3057)} = -5.9432$, $p < 0.001$), Daejeon ($t_{(3084)} = -7.1834$, $p < 0.001$), Ulsan ($t_{(2384)} = -3.7897$, $p < 0.001$). Except for Daegu, spatial accessibility by walking was greater than that by driving in cities.

### 3.3. Intra-Urban Disparity of Access

To explore the inequality in spatial accessibility to primary care for aging populations across the major cities in South Korea, we used the Gini coefficient (Table 2). In terms of inequality in spatial accessibility by driving, the magnitude is not noticeably different across cities. In other words, the aging population living in all cities is unlikely to experience differential access to primary care if they are driving. On the other hand, the inequality in spatial accessibility by walking is greater. This indicates varying levels of accessibility to primary care. As shown in Figure 3, there are fewer areas with adequate spatial accessibility by walking than by driving. Interestingly, as shown in Table 2, Daegu has a higher driving Gini coefficient and a lower walking Gini coefficient, which implies that accessibility to primary care for older adults in Daegu would be improved when public transit services are largely implemented.

**Table 2.** Gini Coefficients of Spatial Accessibility in the Older Population.

| City | Gini Coefficient | |
|---|---|---|
| | Driving | Walking |
| Seoul | 0.0844 | 0.555 |
| Busan | 0.0458 | 0.5491 |
| Incheon | 0.0045 | 0.5954 |
| Daegu | 0.0834 | 0.5453 |
| Gwangju | 0.022 | 0.5687 |
| Daejeon | 0.0139 | 0.6598 |
| Ulsan | 0.0904 | 0.6686 |

The correlation between the aging index and spatial accessibility by city helps provide an improved understanding of how primary care services are distributed in predominantly older regions (Table 3). Positive coefficients indicate that the spatial distributions of those two variables were well-matched. However, negative values of the correlation coefficient signify that elderly populations have less access to primary care than other age groups. Put differently, primary care services are mainly located in regions in which residents are younger (i.e., less than 65 years old).

**Table 3.** Correlation between the aging index and healthcare accessibility.

| City | Kendall's Tau | |
|---|---|---|
| | Driving | Walking |
| Seoul | 0.0621 *** | 0.017 *** |
| Busan | 0.1182 *** | 0.004 |
| Incheon | 0.2492 *** | −0.0753 *** |
| Daegu | 0.117 *** | −0.1529 *** |
| Gwangju | 0.0864 *** | −0.0968 *** |
| Daejeon | 0.0893 *** | −0.1702 *** |
| Ulsan | 0.0991 *** | −0.0605 *** |

$p$-value < 0.001 ***.

There are positive correlation coefficients between accessibility by driving and the aging index. This suggests that aging populations would be well-served by nearby healthcare resources if they had access to cars. However, negative coefficients were detected in the correlation between healthcare access by walking and the aging index. This means that the major cities (other than Seoul and Busan) did not serve their aging populations

well. In other words, these results indicate that more healthcare services are needed to improve accessibility among the aging population, and this is especially critical for the cities of Incheon, Daegu, Gwangju, Daejeon, and Ulsan.

## 4. Conclusions

Our study investigated the spatial accessibility to primary care for aging populations in seven major cities in South Korea. Using the E2SFCA method, we measured the spatial accessibility within 30-min driving and 15-min walking travel times, considering distance decay (i.e., how people are more likely to visit a primary care provider closer to their home). The measures of spatial accessibility clearly delineate the locations of higher accessibility from those suffering from insufficient access within each city. City planners can utilize our results to inform the future planning and development of their cities and to improve older residents' access to healthcare, which would, in turn, enhance population health and well-being.

By investigating inter- and intra-urban patterns in South Korea, we expand on well-known urban-rural inequities in healthcare access [20,21] by recognizing the spatial heterogeneity within urban areas. We uncover inter- and intra-urban variability in spatial accessibility to primary care for aging populations that would have been overlooked had we utilized the urban/rural binary to categorize our study area. In doing so, we are able to identify which cities and neighborhoods have the highest and lowest levels of spatial accessibility.

We uncover two main findings related to intraurban and interurban patterns. First, older populations living in the suburbs and peripheral areas of cities have the worst spatial access to primary care, while those living in the city centers have better spatial access. Second, while the spatial accessibility of all cities could use improvement, Seoul emerged as having the highest healthcare spatial accessibility, whereas Busan and Ulsan were identified as the cities most in need of additional primary care services. In addition, many areas in Incheon have alarmingly poor spatial accessibility to primary care.

The spatial accessibility measures imply that more healthcare professionals need to be placed in the suburban and peripheral areas of cities. The issue is that healthcare professionals may not prefer to work there. To address this critical problem, the South Korean government established a policy that male medical doctors should serve as public health doctors instead of their mandatory military service. They are sometimes required to work in island or rural areas (e.g., areas that have poor healthcare accessibility). In spite of such policies, it may still be insufficient to improve the accessibility of the aging populations residing in islands and the peripheral areas of cities. Recently, there have been debates about whether public medical schools should be established in rural areas [29]. Mobile healthcare vans and boats may be useful in addressing such issues. Placing new hospitals, clinics, or medical schools require financial resources, perhaps beyond what cities can afford to spend. Instead, more flexible and economical services such as mobile healthcare vans and boats would help.

This study has limitations. First, we did not consider temporal dynamics in our measures of spatial accessibility. Spatial accessibility may vary over time [25]. The number of residents in each region also varies over time due to human mobility. Hospitals may not be open 24 h a day and seven days a week. Traffic may also impede people's access to hospitals. Second, the assessment results measured by the E2SFCA would be improved once a historical hospital visit record is available and taken into consideration to reflect travel behavior or preference [30]. Individual-based [31] or collective accessibility [32] may be suitable for evaluating spatial accessibility to healthcare services considering both space and time. Future research endeavors should include the collection and analysis of data on human mobility and temporal constraints to more accurately model spatial access to healthcare services at finer temporal scales. Not only that, leveraging the information about the car ownership rate of the aging population would help to accurately assess the spatial access to the primary cares. Although it is not easily obtained the information about the car



ownership rates in South Korea, it would be more interesting to investigate to what extent the spatial access to primary care varies, considering the uncertainty about car ownership.

Investigations into healthcare access among older adults are urgently needed around the world as many countries experience a rapidly growing aging population in tandem with increased utilization of health services. While our project focuses on South Korea and provides much-needed insights there, we utilize methods that are easy to implement elsewhere. More research on this topic will help identify areas in critical need of resources as well as possibilities for improving healthcare access among older populations.

**Author Contributions:** Conceptualization, Jeon-Young Kang and Sandy Wong; Methodology, Jeon-Young Kang, Sandy Wong and Jinwoo Park; Formal analysis, Jeon-Young Kang, Jinwoo Park and Jinhyung Lee; Writing—original draft, Jeon-Young Kang; Writing—review & editing, Jared Aldstadt. All authors have read and agreed to the published version of the manuscript.

**Funding:** This research received no external funding.

**Data Availability Statement:** The data that support the findings of this study are available from the first author (Kang, J.-Y.), upon reasonable request.

**Conflicts of Interest:** The authors declare no conflict of interest.

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
