# Peer review of "Exploring Spatial Mismatch between Primary Care and Older Populations in an Aging Country: A Case Study of South Korea"

_ijgi, doi:10.3390/ijgi12070255_

Round 1
Reviewer 1 Report
This is a well-written article evaluating primary accessibility for the elderly in seven Korean cities. The paper employs the enhanced 2SFCA model for the evaluation and then correlated access with the proportion of elderly to reveal inequality issues. Below are my major concerns.
- I think there should be a summary/introduction about how the seven cities are selected and their differences in gerontology policy and issues. This background information could set the tone for the discussion on regional differences later in the article.
- I didn’t expect there to be a discussion on travel mode (walking vs. driving) until the method. The author should introduce some literature on travel mode in the first place, as I know there have been numerous studies about travel modes in 2SFCA and its extensions. Also, why are the three travel time zones (L177) the same for walking and driving? I believe the specification of travel time zones in the original E2SFCA [10] is for driving only. Last, how do you derive the walking and driving time in the road network? I think there are some method justifications and specifications to fill in.
- It seems to me that the numbers in Table 3 could be wrong. For example, for Seoul, the driving average is 0.0144 with the 95% CI being the same numbers (0.0144 - 0.0144). Please double-check the results.
- In the last method section, you have a correlation analysis between accessibility and the elderly population. However, the elderly population (X) is a variable in modeling the accessibility (Y), meaning that the two variables are somewhat related in the first place.
Minor issues:
- L61: “address the underlying assumptions” doesn’t sound right.
- L66: “In our paper, we chose to utilize the E2SFCAM because it is intuitive, clearly demarcates accessibility shortage areas, and is easy to implement.” This sentence can be improved grammatically.
- L73: “A good example is Yun, Kim [18].” The in-text citation format doesn’t seem correct. There are other formatting issues regarding in-text citations throughout the article.
- L103: “To measure the elderly population’s spatial accessibility to primary care locations, we use the Enhanced Two-Step Floating Catchment Area Method (E2SFCAM).” This sentence has been repetitive.
- L148: Please add some explanation to dongs, as it is not a familiar concept (at least to non-Korean readers).
- It is by convention that the two-step floating catchment area method is abbreviated as the 2SFCA method, instead of 2SFCAM as in the paper. In the literature review (L49-68) and also some other sections, the paper has added that extra M to the different 2SFCA methods. Please consider dropping the M.
- Table is not necessary. Simply describing the included fields in a sentence is good enough.
- You need to explain the E2SFCA equation (L177) and how it differs from the basic 2SFCA.
- The description of the variables and their positions in the equation (L177) is not clear. For example, L173: “followed Luo and Qi [10]’s methods to apply the values 1, 0.68, and 0.22 to the 0-10, 10-20, and 20-30 travel time zones.”
- L206: “Notably, from PPR to average spatial accessibility, Incheon drops from fourth place to last while Busan elevates from last place to fifth.” Walking? Driving? Are walking rank and driving rank the same across cities?
- L217: there should always be a space before and after <. Please check throughout the article. This formatting issue permeates through the paper.
- L241: Gini coefficient needs some explanation.
- L262. The Spearman correlation coefficients need some explanation.
Author Response
This is a well-written article evaluating primary accessibility for the elderly in seven Korean cities. The paper employs the enhanced 2SFCA model for the evaluation and then correlated access with the proportion of elderly to reveal inequality issues. Below are my major concerns.
Authors: Thank you for your insightful reviews. We have thoroughly revised our manuscript based on your comments.
- I think there should be a summary/introduction about how the seven cities are selected and their differences in gerontology policy and issues. This background information could set the tone for the discussion on regional differences later in the article.
Authors: Thank you. We have added the following information in the last paragraph of the introduction.
“Our study investigates healthcare spatial accessibility for older populations (aged 65 or over) and its disparity for intra- and inter-urban areas, focusing on the seven major cities in South Korea (i.e., Seoul, Busan, Incheon, Daegu, Daejeon, Gwangju, and Ulsan). While they have a relatively less percentage of elderly people now, it is projected that the proportion will significantly increase in twenty years due to the aging population in South Korea. These cities were selected as their metropolitan governments have autonomous rights for decision-making of infrastructure allocation.”
In addition, the following sentence is added in Section 2.1. Study Area.
“In South Korea, many rural areas have already become the super-aged societies. More importantly, the seven major cities will also become the super-aged in the near future [26].”
- I didn’t expect there to be a discussion on travel mode (walking vs. driving) until the method. The author should introduce some literature on travel mode in the first place, as I know there have been numerous studies about travel modes in 2SFCA and its extensions. Also, why are the three travel time zones (L177) the same for walking and driving? I believe the specification of travel time zones in the original E2SFCA [10] is for driving only. Last, how do you derive the walking and driving time in the road network? I think there are some method justifications and specifications to fill in.
Authors: Thank you for catching our mistake. We have added the following sentences in the last paragraph of the introduction.
“First, we measured spatial accessibility to healthcare resources for the older population in the seven cities. This step employed the Enhanced Two-Step Floating Catchment Area (E2SFCA) method [9], given that it is widely implemented in healthcare accessibility studies [23, 24] as well as its intuitive characteristics [25]. In addition, we explored different accessibility attributed to different transportation modes (i.e., driving and walking).”
In addition, we have supplemented the information on how we retrieved road network datasets for walking and driving, as shown below (see the last paragraph of section 2.2. Data)
“Using OSMnx, we retrieved two different road network datasets, calculating travel time for vehicles and walks. As our purpose of this study is not only to analyze the spatial accessibility to primary care for older populations but also to compare the spatial accessibility measures via driving and those via walks, we downloaded both road network datasets and utilized them.”
- It seems to me that the numbers in Table 3 could be wrong. For example, for Seoul, the driving average is 0.0144 with the 95% CI being the same numbers (0.0144 - 0.0144). Please double-check the results.
Authors: Thank you for the comment. We have removed the table, and moved the contents into the first paragraph of Section 3.2. Inter-urban disparity of access.
- In the last method section, you have a correlation analysis between accessibility and the elderly population. However, the elderly population (X) is a variable in modeling the accessibility (Y), meaning that the two variables are somewhat related in the first place.
Authors: Thank you for the insightful comment. We have replaced the X variable of the correlation from the elderly proportion to the aging index (population over 65 years old / population less than 15 years old). You can see the detailed information on the correlation analysis in the last paragraph of Section 2.3. Method.
“Furthermore, Kendall’s tau correlation is employed to assess the mismatch between primary care spatial accessibility and the aging index per city. Given that we employed the aging index instead of the aging population counts as a correlation variable, it is considered to be exempt from the potential concern of collinearity.”
Minor issues:
- L61: “address the underlying assumptions” doesn’t sound right.
Authors: the sentence is revised as follows. We have used the word “assumption” since it was mentioned in the cited article.
“The M2SFCA employs an additional distance decay function only for supply to address the assumptions of 2SFCA metrics (i.e., every supply location is optimal for serving demand).”
- L66: “In our paper, we chose to utilize the E2SFCAM because it is intuitive, clearly demarcates accessibility shortage areas, and is easy to implement.” This sentence can be improved grammatically.
Authors: We have updated the sentence as follows.
“This step employed the Enhanced Two-Step Floating Catchment Area (E2SFCA) method [9], given that it is widely implemented in healthcare accessibility studies [23, 24] as well as its intuitive characteristics [25].”
- L73: “A good example is Yun, Kim [18].” The in-text citation format doesn’t seem correct. There are other formatting issues regarding in-text citations throughout the article.
Authors:
- L103: “To measure the elderly population’s spatial accessibility to primary care locations, we use the Enhanced Two-Step Floating Catchment Area Method (E2SFCAM).” This sentence has been repetitive.
Authors: Thank you. We revised the entire manuscript to avoid the issue.
- L148: Please add some explanation to dongs, as it is not a familiar concept (at least to non-Korean readers).
Authors: We have updated the sentence as shown below, explaining “dong” for non-Korean readers.
“We also extracted demographic data at the scale of dongs, which are administrative units that are proxies for neighborhoods.”
- It is by convention that the two-step floating catchment area method is abbreviated as the 2SFCA method, instead of 2SFCAM as in the paper. In the literature review (L49-68) and also some other sections, the paper has added that extra M to the different 2SFCA methods. Please consider dropping the M.
Authors: We removed every M from the words containing 2SFCAM.
- Table is not necessary. Simply describing the included fields in a sentence is good enough.
Authors: We have removed two tables (Hospital dataset schema and Averaged spatial accessibility scores) to make the manuscript concise.
- You need to explain the E2SFCA equation (L177) and how it differs from the basic 2SFCA.
Authors: We have added the following information to the fourth paragraph of the introduction.
“The original 2SFCA method considers whether a location is accessible to a particular service of interest in a Boolean manner (i.e., accessible or not accessible). To resolve this limitation of the conventional approach, there have been advanced developments on the 2SFCA method, … ”
- The description of the variables and their positions in the equation (L177) is not clear. For example, L173: “followed Luo and Qi [10]’s methods to apply the values 1, 0.68, and 0.22 to the 0-10, 10-20, and 20-30 travel time zones.”
Authors: We have updated the section as shown below.
“In order to take into account the distance decay effect (i.e., people may be more likely to visit relatively closer primary care hospitals than further ones), we set up three travel time zones: 0-10, 10-20, and 20-30 minutes for driving and 0-5, 5-10, and 10-15 minutes for walking. For each travel time zone, we followed Luo and Qi [9]’s approach and applied the weights of 1, 0.68, and 0.22 to each of the three travel time zones (e.g., a weight of 0.68 for 10-20 driving time and 5-10 walking time), respectively. Then we made our final calculations using the following formula:”
- L206: “Notably, from PPR to average spatial accessibility, Incheon drops from fourth place to last while Busan elevates from last place to fifth.” Walking? Driving? Are walking rank and driving rank the same across cities?
Authors: We provided detailed information about why and how the accessibility varies between cities per mobility, as shown below.
Spatial accessibility by driving provided a relatively uniform distribution compared to the ones by walking (Figure 4). Seoul has the most uniform distribution, while Daegu has the most discrete distribution. Seoul is the most developed city in South Korea, many infrastructures have been concentrated in Seoul. This may be responsible for the relatively even distribution of spatial accessibility in Seoul. It is interesting that Daegu has three groups in its accessibility histogram. This pattern echoes the spatial distribution of accessibility (Figure 3). While the southeast part of the city has beyond sufficient accessibility, the southwest and west part of the city has very insufficient accessibility.
Spatial accessibility by walking provided right-skewed distribution for every city. This indicates that people in many subregions within the city are less accessible to primary care. In other words, strong spatial disparities in healthcare accessibility are detected. Compared to its uniform distribution of driving accessibility, Seoul has a very high accessibility score (799.85) as its maximum value and provides 0 as its minimum accessibility score. In summary, spatial accessibility provides a very distinctive pattern between different modalities (i.e., driving and walking).
- L217: there should always be a space before and after <. Please check throughout the article. This formatting issue permeates through the paper.
Authors: We have fixed the issue.
- L241: Gini coefficient needs some explanation.
- L262. The Spearman correlation coefficients need some explanation.
Authors: We provided the following information in the last paragraph of Section 2.3. Method to articulate Gini coefficients and correlation analysis.
“In addition, the Gini coefficients explore spatial inequalities in the geographical access to primary care within each major city. Among works of literature, the Gini has already been advocated as a useful indicator for measuring spatial inequality of spatial accessibility[14, 27]. Furthermore, Kendall’s tau correlation is employed to assess the mismatch between primary care spatial accessibility and the aging index (i.e., population over 65 years old / population less than 15 years old) per city. Given that we employed the aging index instead of the aging population counts as a correlation variable, it is considered to be exempt from the potential concern of collinearity.”
Reviewer 2 Report
An excellent example of spatial accessibility analysis for several large cities in South Korea. I think this study was well designed and the manuscript is well written.
Line 191 - You mention the color red in Figure 3, but Figure 3 uses shades of green.
Line 235 - This confused me. How is spatial accessibility higher by walking than driving? I'd like to see this addressed more.
Author Response
Reviewer 2
An excellent example of spatial accessibility analysis for several large cities in South Korea. I think this study was well designed and the manuscript is well written.
Authors: Thank you for your positive review and your time reviewing our manuscript.
Line 191 - You mention the color red in Figure 3, but Figure 3 uses shades of green.
Authors: We fixed the typo.
Line 235 - This confused me. How is spatial accessibility higher by walking than driving? I'd like to see this addressed more.
Authors: We provided detailed information about why and how the accessibility varies between cities per mobility, as shown below.
Spatial accessibility by driving provided a relatively uniform distribution compared to the ones by walking (Figure 4). Seoul has the most uniform distribution, while Daegu has the most discrete distribution. Seoul is the most developed city in South Korea, many infrastructures have been concentrated in Seoul. This may be responsible for the relatively even distribution of spatial accessibility in Seoul. It is interesting that Daegu has three groups in its accessibility histogram. This pattern echoes the spatial distribution of accessibility (Figure 3). While the southeast part of the city has beyond sufficient accessibility, the southwest and west part of the city has very insufficient accessibility.
Spatial accessibility by walking provided right-skewed distribution for every city. This indicates that people in many subregions within the city are less accessible to primary care. In other words, strong spatial disparities in healthcare accessibility are detected. Compared to its uniform distribution of driving accessibility, Seoul has a very high accessibility score (799.85) as its maximum value and provides 0 as its minimum accessibility score. In summary, spatial accessibility provides a very distinctive pattern between different modalities (i.e., driving and walking).
Reviewer 3 Report
Well-constructed study addressing spatial accessibility to providers by static location. My one comment is regarding Figure 1 - for panel (A), is this the number of elderly per 10,000? As it reads now, the population proportion would already be nearly 100%.
Author Response
Well-constructed study addressing spatial accessibility to providers by static location. My one comment is regarding Figure 1 - for panel (A), is this the number of elderly per 10,000? As it reads now, the population proportion would already be nearly 100%.
Authors: Thank you for your positive review and your time reviewing our manuscript. We need to remove ‘per’ from Figure 1.
Round 2
Reviewer 1 Report
The paper has been improved. The structure is much better now. The result showing the different levels of Gini indices by car and walking is interesting. I am particularly interested in the conclusion that “It is indicated that elderly populations would be well-served by surrounding healthcare resources if they could have access to cars.” The authors may want to expand or further discuss the result. For example, what is the current car ownership rates among the elderly in Korea as whole and in different cities? Which city could be largely benefited from transport planning solutions to improve medical access (e.g., improving public transit would be most effective for a city with a high driving Gini but low walking Gini)?
However, the paper still suffers from numerous writing issues, which must be addressed. The paper needs proofread. Please have a professional editor proofread the paper. If writing issues persist in the next version, I will refrain from further reviewing the article.
Abstract: “The regions with more senior populations have lower accessibility to primary care.” I believe you used the aging index which represents the ratio rather than the absolute senior populations, so this sentence is not precise.
There are minor grammatical glitches throughout the article. For example, “in the urban planning perspective (L42)” should be “from the perspective” or “in a perspective of xxx.” “it puts more efforts on,” who is “it”? “The M2SFCA employs an additional distance decay function only for supply to address the assumptions of 2SFCA metrics (i.e., every supply location is optimal for serving demand).” You cannot address the assumptions; you can only address issues or validate assumptions.
L75: “much attention has been paid to the urban scholars and public health scientists” Attention is paid to scientists?
L77: “A good example is [17]”. Author name(s) is needed for such citation, such as “A good example is XXX et al. [17], which xxx.” The same applies to many other citations, such as [18], and later on [9]. Please check through the article.
L116: E2SFCA only needs full term on its first use.
L192: Why the “0-10, 10-20, and 20-30 minutes for driving and 0-5, 5-10, and 10-15 minutes for walking” are selected? Justification is needed.
L206: ANOVA should be in the () instead.
L208: indicate -> can indicate
L214: aging index needs references.
L278: You need to justify the use of the Gini index, such as citing this paper, which uses the Gini index to evaluate the inequality of school accessibility by 2SFCA: “Improving educational equity by maximizing service coverage in rural Changyuan, China: An evaluation-optimization-validation framework based on spatial accessibility to schools.”
Author Response
Reviewer 1 Comments
Comment 1-1: The paper has been improved. The structure is much better now. The result showing the different levels of Gini indices by car and walking is interesting. I am particularly interested in the conclusion that “It is indicated that elderly populations would be well-served by surrounding healthcare resources if they could have access to cars.” The authors may want to expand or further discuss the result. For example, what is the current car ownership rates among the elderly in Korea as whole and in different cities?
Response 1-1: Thank you for your suggestion. In our analysis, we assumed that older adults would take a cab, rather than public transits. Unfortunately, there are no official datasets about car ownership in South Korea. At least from my understanding, Shin (2021) finds that about 70% of older adults (over 60 years) owned the vehicle. So, it would be more interesting to investigate to what extent the spatial access to the primary cares is varying considering the uncertainty about the car ownership. As you suggested, we have expanded our conclusion containing the explanations about the car ownership or so.
Shin, E. J. (2021). Exploring the causal impact of transit fare exemptions on older adults’ travel behavior: Evidence from the Seoul metropolitan area. Transportation Research Part A: Policy and Practice, 149, 319-338.
Comment 1-2: Which city could be largely benefited from transport planning solutions to improve medical access (e.g., improving public transit would be most effective for a city with a high driving Gini but low walking Gini)?
Response 1-2: Thank you for your help to highlight the results from our analysis. As shown in Table 2, Daegu has a higher driving Gini coefficient and lower walking Gini coefficient, it implies that an accessibility to the primary cares for older adults in Daegu would be improved when public transit services are more placed.
Comment 1-3: However, the paper still suffers from numerous writing issues, which must be addressed. The paper needs proofread. Please have a professional editor proofread the paper. If writing issues persist in the next version, I will refrain from further reviewing the article.
Response 1-3: Thank you for your suggestion. We have reviewed the manuscript and corrected all writing issues.
Comment 1-4: Abstract: “The regions with more senior populations have lower accessibility to primary care.” I believe you used the aging index which represents the ratio rather than the absolute senior populations, so this sentence is not precise.
Response 1-5: Thank you for your comment. We have revised the sentence.
Comment 1-5: There are minor grammatical glitches throughout the article. For example, “in the urban planning perspective (L42)” should be “from the perspective” or “in a perspective of xxx.” “it puts more efforts on,” who is “it”? “The M2SFCA employs an additional distance decay function only for supply to address the assumptions of 2SFCA metrics (i.e., every supply location is optimal for serving demand).” You cannot address the assumptions; you can only address issues or validate assumptions.
L75: “much attention has been paid to the urban scholars and public health scientists” Attention is paid to scientists?
L77: “A good example is [17]”. Author name(s) is needed for such citation, such as “A good example is XXX et al. [17], which xxx.” The same applies to many other citations, such as [18], and later on [9]. Please check through the article.
L116: E2SFCA only needs full term on its first use.
L192: Why the “0-10, 10-20, and 20-30 minutes for driving and 0-5, 5-10, and 10-15 minutes for walking” are selected? Justification is needed.
L206: ANOVA should be in the () instead.
L208: indicate -> can indicate
L214: aging index needs references.
Response 1-5: Thank you for your comments on the grammar issue. The specific grammatical errors noted here have been corrected. Other grammatical errors have been corrected throughout the manuscript.
Comment 1-6: L278: You need to justify the use of the Gini index, such as citing this paper, which uses the Gini index to evaluate the inequality of school accessibility by 2SFCA: “Improving educational equity by maximizing service coverage in rural Changyuan, China: An evaluation-optimization-validation framework based on spatial accessibility to schools.”
Response 1-6: We have cited the paper, titled Improving educational equity by maximizing service coverage in rural Changyuan, China: An evaluation-optimization-validation framework based on spatial accessibility to schools.
Han, Z., Cui, C., Kong, Y., Li, Q., Chen, Y., & Chen, X. (2023). Improving educational equity by maximizing service coverage in rural Changyuan, China: an evaluation-optimization-validation framework based on spatial accessibility to schools. Applied Geography, 152, 102891.
